# Immunoglobulin Disorders and the Oral Cavity: A Narrative Review

**DOI:** 10.3390/jcm11164873

**Published:** 2022-08-19

**Authors:** Maja Ptasiewicz, Dominika Bębnowska, Paulina Małkowska, Olga Sierawska, Agata Poniewierska-Baran, Rafał Hrynkiewicz, Paulina Niedźwiedzka-Rystwej, Ewelina Grywalska, Renata Chałas

**Affiliations:** 1Department of Oral Medicine, Medical University of Lublin, 20-093 Lublin, Poland; 2Institute of Biology, University of Szczecin, 71-412 Szczecin, Poland; 3Doctoral School, University of Szczecin, 71-412 Szczecin, Poland; 4Department of Experimental Immunology, Medical University of Lublin, 20-093 Lublin, Poland

**Keywords:** oral mucosa, immunoglobulins, mucous membrane pemphigoid (MMP), pemphigus vulgaris (PV), linear IgA bullous dermatosis (LABD), Epidermolysis Bullosa Aquisita (EBA), Hyper-IgE Syndrome (HIES)

## Abstract

The oral mucosa is a mechanical barrier against the penetration and colonization of microorganisms. Oral homeostasis is maintained by congenital and adaptive systems in conjunction with normal oral flora and an intact oral mucosa. Components contributing to the defense of the oral cavity include the salivary glands, innate antimicrobial proteins of saliva, plasma proteins, circulating white blood cells, keratinocyte products of the oral mucosa, and gingival crevicular fluid. General disturbances in the level of immunoglobulins in the human body may be manifested as pathological lesions in the oral mucosa. Symptoms of immunoglobulin-related general diseases such as mucous membrane pemphigoid (MMP), pemphigus vulgaris (PV), linear IgA bullous dermatosis (LABD), Epidermolysis Bullosa Aquisita (EBA), and Hyper-IgE syndrome (HIES) may appear in the oral cavity. In this review, authors present selected diseases associated with immunoglobulins in which the lesions appear in the oral cavity. Early detection and treatment of autoimmune diseases, sometimes showing a severe evolution (e.g., PV), allow the control of their dissemination and involvement of skin or other body organs. Immunoglobulin disorders with oral manifestations are not common, but knowledge, differentiation and diagnosis are essential for proper treatment.

## 1. Introduction

The oral mucosa is constantly exposed to multiple triggers that require immune control, such as bacteria, allergens and local damage caused by chewing food [1]. It is a mechanical barrier limiting the invasion by commensal and pathogenic microorganisms, also inhabited by opportunistic microorganisms that are involved in the development of not only oral diseases, but also some systemic diseases [2]. Oral homeostasis is maintained by congenital and adaptive systems in conjunction with normal oral flora and an intact oral mucosa. Components contributing to the defense of the oral cavity include the salivary glands [3], innate antimicrobial proteins of saliva [4], plasma proteins, circulating white blood cells [5], keratinocyte products of the oral mucosa [6] and gingival crevicular fluid [7,8]. The oral cavity is the place where first symptoms of systemic immune diseases may appear. Often, signs of general diseases are seen in the mouth even before the underlying disorder is diagnosed, and they can expedite diagnosis and provide a chance for prompt treatment. Periodontal diseases affects not only local lesions in the oral cavity, but there is also a possible relationship between periodontitis and systemic or autoimmune diseases [2,9], which has been widely reported, e.g., in diabetes [10], inflammatory bowel disease [11,12], systemic lupus erythematosus [13], subsequent Sjögren’s syndrome [14], psoriasis [15], rheumatoid arthritis [16], cardiovascular disease [17] and chronic kidney disease [18].

Below, we present selected immunoglobulin defects’ detrimental manifestations on the periodontal and other oral tissues.

## 2. Salivary Immunoglobulins

Saliva contains mucins, enzymes, and secretory immunoglobulin A (sIgA) that protect the oral mucosa against microorganism colonization [19]. The gingival crevicular fluid contains leukocytes, IgG, IgM, IgA, and other factors that contribute to oral immunity [20]. Each immunoglobulin has different chemical and biological properties. There are two major classes of antibodies present in human saliva: secretory IgA (sIgA) and IgG. Dimeric IgA is produced by plasma cells in the stroma of the salivary glands and then transported through the glandular epithelial cells by the polymeric Ig receptor or the membrane secretory component. On the apical surface of the epithelial cell, sIgA is exocytosed after cleavage of the Ig receptor [21]. The monomeric (non-secretory) fraction of IgA in the saliva is small (13–17% of total IgA saliva) and enters the oral cavity mainly through the gingival fluid or mucosal exudate [22]. As patients with inflammatory bowel disease (IBD) have an increased secretion of IgA, elevated serum levels of this immunoglobulin may indicate existing inflammation throughout the gastrointestinal tract, not just in the oral cavity [23,24]. IgG constitutes 75% of the serum immunoglobulins and is the major component of the secondary antibody response [25]. IgG is also the immunoglobulin that crosses the placenta, giving protection to the newborn [26]. A major part of salivary IgG comes from the blood by passive leakage through the gingival fluid, and only a small part comes from the salivary glands [21].

IgM, IgD, and IgE fractions in saliva are small and are mainly derived from gingival leakage, and salivary IgM levels have been correlated with both serum IgM and periodontitis [27]. IgM antibodies are macromolecules composed of five antibody monomers and are produced chiefly during the body’s primary response to a foreign antigen [28]. IgM also plays an important role in the activation of complement and in the formation of immune complexes, which are aggregates of antibodies, antigens, and complements [21]. Most IgM in the body is intravascular.

## 3. Oral Mucosa Immunity

The oral mucosa contains a similar amount and diversity of microbiota as the lower bowel, with more than 700 different species of bacteria, viruses, and fungi, both commensal and opportunistic [29,30]. As in the entire human body, also in the oral cavity, the first line of defense against pathogens is the oral epithelium, which primarily acts to neutralize foreign and harmful antigens and prevents colonization of the oral cavity by pathogenic microorganisms [31,32]. This is made possible by the action of many immune cells present there, which have the ability to secrete a variety of immunosuppressive mediators [31]. The primary role of the squamous epithelium in the oral cavity is to maintain a mechanical barrier to microorganisms. To do this, a number of mechanisms have evolved. First, the exfoliation of the epithelial scales is designed to limit colonization by epithelial microorganisms. Furthermore, antibodies produced by B lymphocytes, Langerhans cells that present antigen to helper T lymphocytes, and specific cytokines provide another barrier against unwanted microorganisms [33]. Immune components unique to the oral mucosa include saliva secretory immunoglobulin A (sIgA), gingival fluid components, and keratinocyte secretion of biological mediators by keratinocytes [31].

An important element that plays a crucial role in the balance of the oral cavity is saliva, which is composed of a wide variety of antimicrobial components such as IgA, IgM and IgG immunoglobulins; histatins; lysozyme; lactoferrin; and peroxidases [32]. The importance of saliva in antimicrobial defense processes is indicated by studies of patients with decreased or absent salivary secretion, who were more likely to have oral candidiasis [34]. The site in the oral cavity where most inflammatory cells occur is the gingiva due to its frequent mechanical damage caused by chewing and hygiene, and the large and diverse microbiota inhabiting the teeth [32].

Pattern recognition receptors (PRRs) have the ability to distinguish pathogen-associated molecular patterns (PAMPs) from their own antigens, thus producing an immune and inflammatory response only against non-self-antigens [31]. PRRs on the surface of dendritic cells (DC) are involved in the adaptive immune response by presenting the detected antigen to T cells with an immature CD83+ phenotype [35]. A study by Jotwani et al. shows a significant role of DC in chronic periodontitis (CP) [29]. Among DCs in the oral epithelium, Langerhans cells (LCs) should be distinguished. LCs maintain a state of immune tolerance and limit the activity of effector T cells by secreting anti-inflammatory cytokines, interleukin-10, and transforming growth factor b1. PRRs—CD1a, Fc 3RI, CD11b and langerin/CD207—are expressed on their surface [31]. Toll-like receptors (TLRs) are other PRRs that act in oral immunity. TLRs distinguish host components from pathogens through pathogen-associated molecular patterns (PAMPs), which is particularly important in the oral cavity because of the constant contact between oral microbes and the host immune system. As a result of contact of TLR with PAMP, intracellular signaling pathways are stimulated that lead to immune cell activation and the production of cytokines and defensins [36]. TLRs are expressed mainly in neutrophils, monocytes/macrophages, and dendritic cells, among others, i.e., immune cells belonging to the innate immune system, but have recently also been found in B and T lymphocytes (cells of acquired immunity) [36]. In gingival epithelial cells, TLR-2-6 and -9 are expressed. Stimulated by pathogenic microorganisms, TLRs activate the release of the antimicrobial β-defensins cathelicidin and calprotectin, and interleukin 8, which is described as a neutrophil chemoattractant [36]. Due to periodontal exposure to many commensal and pathogenic microorganisms and high stimulation by PAMPs, TLRs can overproduce proinflammatory cytokines [36]. The result of overproduction can be the destruction of the epithelial barrier and even connective tissue and bone [37]. A study by Schröder et al. [38] showed that TLR-4 is such an important part of chronic periodontitis that it may be a risk factor for the development of this disease [38]. In a periodontitis model in C3H/HeJ mice, reduced bone respiration was observed when TLR-4 deficiency prevailed [39].

The function of neutrophils in oral and periodontal immunity includes not only microbial killing, but more importantly regulates environmental homeostasis by preventing the development of periodontal immunopathology in the patient [32]. Neutrophils maintain regulation in IL-17/Th17 response [32]. However, to maintain health, neutrophil counts must not be too low or too high, as this can lead to the development of immunopathology in the oral cavity [32]. T lymphocytes, B lymphocytes, and innate lymphoid cells (ILC) are found in the gums. Among these cells, CD4+ T lymphocytes have been identified as mediators of periodontitis pathology. Th17 cells also play an important role in maintaining balance. The number of Th17 cells in the gingiva increases with increasing patient age, regardless of microbial load, distinguishing Th17 gingival cells from Th17 skin and intestinal cells [32].

Cytokines and chemokines secrete signals that regulate the initiation of the immune response to pathogens [30]. Among the cytokines secreted in the oral cavity, the most relevant are interleukin-1 beta, interleukin 6, tumor necrosis factor alpha, granulocyte-macrophage colony-stimulating factor (GM-CSF), transforming growth factor beta (TGF-beta) and their receptors, and interleukin-8 (IL-8) [33]. IL-8 is a major neutrophil polymorphonuclear (PMN) chemoattractant and is secreted primarily by gingival keratinocytes [30]. CXCL10 is secreted in response to interferon-γ, tumor necrosis factor α, and IL-1β by gingival fibroblasts and is a chemoattractant for activated Th1 cells. Other chemokines, such as CCL2, CCL3, CCL4, and CCL5, show chemotacticity for monocytes and lymphocytes [30].

The immune response depends on the type of disease. Due to various factors, immune responses in the oral cavity can be altered and lead to disease development. Thus, environmental factors such as stress and smoking, but also epigenetic and genetic defects, oral microbiota, or chronic diseases (obesity, diabetes, rheumatoid arthritis), in addition to unfavorable pregnancy outcomes, and the level of oral hygiene have an impact on the dysregulation of the immune response [30,37]. Almost half of the population is at risk of developing periodontal disease. In periodontitis, increased levels of the pro-inflammatory cytokines IL-1 and TNF-a are observed, and the expression of TLR-2 and TLR-4 expression is increased [38]. The severity of gingival disease affects the number of LC and interstitial DCs (IDCs) in their epithelium [35].

A summary of the elements involved in oral mucosa immunity is shown in Figure 1.

## 4. Mucosal Humoral Immune Response

The humoral response in the oral cavity is induced by commensal bacteria [31]. Depending on the site, the immune response is different. The main sites of the humoral response are the epithelium, saliva, and gingival crevices [33]. The two main classes of antibodies found in saliva are IgA, which is the product of committed B cells that have undergone a Th cell-dependent or Th cell-independent class switching to IgA, and IgG [40], which were discussed in detail in the previous section. In response to various factors, immune cells migrate to the gums. Neutrophils and leukocytes move to the gingiva in response to plaque, and neutrophils also read IL-8 activity from the gingival epithelium [33]. In a rat model of periodontitis, delivery of the antigen precisely to the gingival mucosa resulted in an efficient humoral immune response [41].

B lymphocytes in periodontal disease can play both a protective and a detrimental role when immunopathology prevails [32]. Because periodontal disease is usually a chronic disease, it is difficult to determine whether persistently high levels of antibodies produced by B lymphocytes protect or contribute to damage [42]. The study by Ebersole et al. [43] showed that although elderly patients probably had a lower quality of humoral response to infections, overall aging had no significant effect on overall oral antibody levels. Among chemokines, CXCL13 is involved in the humoral response as a chemoattractant for B lymphocytes [30].

Attention is drawn to the role of humoral immunity in protecting the oral cavity from candidiasis, especially in patients with a weak immune system, such as those with acquired immunodeficiency syndrome (AIDS) or severe combined immunodeficiency syndrome (SCID) [44]. Furthermore, a case of an immunosuppressed patient was reported that developed *Elizabethkingia miricola* infection associated with periodontitis that caused bacterial translocation [45].

## 5. Oral Manifestations in Selected Diseases

General disturbances in the level of immunoglobulins in the human body may be manifested in pathological lesions in the oral mucosa. Symptoms of immunoglobulin-related general diseases such as pemphigus vulgaris (PV), mucous membrane pemphigoid (MMP), linear IgA bullous dermatosis (LABD), Epidermolysis Bullosa Aquisita (EBA), and Hyper-IgE Syndrome (HIES) may appear in the oral cavity (Table 1).

### 5.1. Mucous Membrane Pemphigoid

Mucous membrane pemphigoid (MMP) is a rare, chronic, subepithelial autoimmune disease that predominantly affects the mucous membranes and, occasionally, skin. It is part of a group of autoimmune dermatoses that present with subepidermal bullous lesions, characterized by the formation of antibodies against structures of the basement membrane zone (BMZ). The majority of cases of MMP IgG directed against antigens on the epidermal side of the skin are detected. In MMP, antibodies bind more frequently to BP-180 and laminin 332 (laminin 5) [46].

MMP is characterized by linear deposition of IgG, IgA, or C3 along the epithelial basement membrane region. In some cases, linear IgE deposits occur as the only immunological complex or additionally for IgG. The current evidence proved that this process develops as a result of a loss of immune tolerance to the structural proteins of the epidermal basal membrane, leading to the development of circulating autoantibodies. The antibodies bind to the basal epidermal membrane, causing inflammation and weakening adhesion of the overlying epidermis [47,48].

Studies have shown that target antigens in the epithelial basement membrane region include pemphigoid 1 (BP230) bullous antigen, pemphigoid 2 bullous antigen (BP180), laminin 5 (α3, β3, γ2 chains), laminin 6 (α3 chain), type VII collagen and the β4 53 integrin subunit BP180 is a transmembrane protein that passes through the laminae and extends into the dense lamina of the basal membrane zone of the epidermis. It is usually the target antigen in approximately 70% of MMP patients [49,50].

MMP generally affects patients over the age of 50 and is twice as common in women than men. The subepithelial lesions of MMP may involve any mucosal surface, but they most frequently involve the oral mucosa [51]. Desquamative gingivitis is the most common manifestation and may be the only manifestation of the disease. MMP is characterized by a gradual onset interspersed with acute exacerbations and remissions. In the majority of patients, the oral mucosa is the site of onset and is the most frequently involved (85%) site, with other sites often affected as well [52]. MMP in the oral mucosa typically manifests as erythematous plaques and erosions that are covered by pseudomembranes, most commonly located on the gingiva and palate and less frequently on the lips, tongue, and cheek mucosa [53]. Scarring of lesions on the mucous membranes is common. A diagnosis of MMP should be considered in patients with bullous lesions or erosions that compromise predominantly mucous membranes [54].

The diagnosis of MMP is based on clinical, histological and immunopathological findings. Clinically, patients must have mainly mucosal disease. Histology shows subepithelial cleavage with a mixed inflammatory infiltrate. Important in the diagnosis is positive direct immunofluorescence (DIF) for IgG, IgA, IgM or C3. It is best to take a sample for histopathological examination from the area of the lesion and another sample from the healthy mucosa, e.g., the cheek [55].

According to the European Guidelines initiated by the European Academy of Dermatology and Venereology, the aim of MMP therapy is to stop inflammation and scarring progression. For oral MMPs, corticosteroids (clobetasol propionate, betamethasone sodium phosphate, fluticasone propionate) or the calcineurin inhibitor tacrolimus are recommended. Clobetasol propionate is used as an ointment on topical lesions, and 0.5 mg betamethasone sodium phosphate diluted in water and fluticasone propionate as an oral rinse. Analgesic, anti-inflammatory and anti-infective drugs may be included in the therapy [55]. The prognosis is usually benign, but the disease should be monitored to ensure that it does not damage other organs. In severe cases, treatment is carried out systemically [56].

### 5.2. Pemphigus Vulgaris (PV)

Pemphigus vulgaris (PV) is one of the oral diseases found to be autoimmune. PV is the most common form of pemphigus, accounting for over 80% of cases. The evolution of PV typically begins with painful mucosal ulceration, especially in the mouth [57]. These ulcers are persistent; individual ulcers may come and go but new lesions continue to appear. Many patients will develop skin lesions over the following weeks or months. PV can affect any site in the oral mucosa with a predilection for gingival papillary tips, in contrast to the diffuse gingival involvement seen in mucous membrane pemphigoid. It is common to see white patches of hyperkeratosis of the mucosa as PV heals [58]. Any mucosal and skin surface may be involved, and in severe cases, the conjunctival, pharyngeal, and laryngeal mucosa may all be involved, along with extensive skin lesions. Patients with oral lesions of pemphigus also frequently have esophageal lesions, and if esophageal symptoms are present, endoscopic examination should be performed to determine the severity of the lesions [59,60]. The underlying mechanism responsible for causing the intraepithelial lesion of PV is the binding of IgG autoantibodies to Dsg1 and DSG 3, a transmembrane glycoprotein adhesion molecule present on desmosomes. It is now known that pemphigus foliaceus (PF) is characterized by autoantibodies to desmoglein (Dsg) 1 and pemphigus vulgaris (PV) is characterized by autoantibodies to Dsg3, although 60% of PV sera also contain Dsg1 autoantibodies [61,62,63]. Importantly, a small number of PV patients may have minimal or no mucosal involvement. These patients typically have relatively low levels of Dsg3 antibodies and higher levels of Dsg1 antibodies. Patients with only mucosal lesions will have no or low levels of Dsg1 antibodies, while patients with the most severe mucocutaneous disease will have high antibody titers to both antigens [64,65].

Mucosal lesions in pemphigus vulgaris (PV) are often followed by skin involvement. Blisters develop within oral mucosa due to the deep intraepidermal cleft between the basal cells and the overlaying spinous keratinocytes. The pathophysiological mechanism causing autoimmune pemphigus is unknown and still being intensively investigated. To date, a catalogue of self-antigens, demonstrated by various authors and detection techniques to react uniquely with pemphigus IgG, includes approximately 20 molecules with different relative molecular masses [66].

The number of target molecules detectable varies from patient to patient and depends on the sensitivity of the detection technique. The technique of immunoblotting versus immunoprecipitation is used. Some of these bands may represent pemphigus antigen degradation products of higher native molecular weights [62]. The detection of pemphigus antigens can be significantly reduced by changing the sensitivity of the technique. This happens when the source of the antigens is first preabsorbed with normal human serum and then used in the PV immunoprecipitation serum. There are only a few protein bands left, including the 85/130 pairs, which were found to be pathophysiologically relevant targets of autoimmunity PV. Similarly, the number of clones detected in the λgt11 keratinocyte Cdna library by PV IgG was reduced by replacing the entire PV IgG fraction with IgG affinity purified single band, the 130 kDa keratinocyte polypeptide [62].

Corticosteroids are recommended for the treatment of PV. Immunosuppressive drugs (prednisolone, azathioprine, cyclophosphamide, cyclosporine and methotrexate) may be included in the therapy. In patients who do not respond to therapies or who have severe disease symptoms, intravenous immunoglobulin (IVIG) is used [56]. In patients with moderate to severe disease, rituximab may also be used. However, in the era of the COVID-19 pandemic, attention is drawn to the careful use of this drug and its possible replacement with IVIG or hydroxychloroquine [67]. A titration of circulating antibodies is performed to determine the progression of the disease, where a high amount of antibodies indicates a severe degree of disease [56].

### 5.3. Linear IgA Bullous Dermatosis (LABD)

Linear IgA bullous dermatosis (LABD) is a rare immune-mediated blistering skin disease with mucosal involvement in some cases. LABD is also known as linear immunoglobulin A dermatosis or linear immunoglobulin A disease and is a relatively rare subepidermal vesiculobullous disease that can occur in both adults and children. LABD is characterized by the deposition of IgA rather than IgG in the basement membrane. IgA anti-basement membrane zone antibodies directed against the 97 kDa portion of BPAG2 (bullous pemphigoid antigen 2) in the lamina lucida are found in this disease. Additionally, some patients may display autoantigens toward LAD-1 (a 120 kDa truncated domain of BPAG2) [66,68]. In most cases, LABD is idiopathic; an association with either drug intake or systemic autoimmune diseases, such as systemic lupus erythematosus, Crohn’s disease, ulcerative colitis, rheumatoid arthritis, and psoriasis has been reported, the latter displaying the most frequent association with LABD [69]. Genetic factors may contribute in part to the development of LABD, and several types of human leukocyte antigens (HLA) are believed to indicate an increased risk of developing the disease. HLA-B8, HLA-DR3, HLA-DQ2, and HLA-cw7 are well-known for their association with LABD [68]. The disease is often drug-induced, with vancomycin being the most frequently charged drug, which is responsible for nearly 50% of drug-induced LABD cases [69]. Additionally, other antibiotics, such as cephalosporins, penicillins, and less often sulfonamide antibiotics, can cause the formation of IgA antibodies, accelerating the development of LABD. Analgesics, antihypertensives, antiepileptics or immunosuppressants have been identified as potential causative agents [69]. Numerous case reports of additional inciting medications are available in the literature, but a few of them include commonly used medications such as allopurinol, amiodarone, furosemide, atorvastatin, glyburide, angiotensin receptor blockers, verapamil, acetaminophen, and the influenza vaccination [68,69].

Oral lesions are common and may affect 50% up to 70% of patients. These changes are clinically indistinguishable from changes occurring in the MMP. Patients develop blisters and erosions of oral mucosa, often accompanied by desquamative gingivitis, that cannot be clinically distinguished from MMP lesions. In some cases, only desquamative gingivitis has been reported. The mucosal involvement in LABD may result in significant scarring [58].

The most important aspect of the diagnosis for LABD is obtaining biopsies for histopathological examination. Additional workup may include indirect immunofluorescence (IIF) to test for the presence or absence of circulating IgA anti-basement membrane zone antibodies, which may be found in up to 70% of LABD patients [68].

For the treatment of LABD, sulfa (dapsone) was recommended as the first line of treatment. However, due to its side-effects, difficult availability and lack of response, rituximab, IVIG, and sulfonamides (sulphapyridine or sulphamethoxypyridazine) in combination with topical and systemic corticosteroids are used more frequently, depending on prognosis. In severe cases and those not responding to therapies, rituximab and IVIG are suggested. In pediatric patients, sulfonamides, topical corticosteroids and nicotinamide are chosen [70,71]. If it is a drug-induced variant of the disease, it should be discontinued as soon as possible [71]. The prognosis of the disease is usually good, with remission occurring and the lesions occurring without scarring. However, the disease should be monitored to ensure that it does not develop into mucosal disease, which progresses to scarring conjunctivitis and can cause blindness [71].

### 5.4. Epidermolysis Bullosa Aquisita (EBA)

EBA is a rare autoimmune blistering disease, which results in vesicle and bullae formation on the skin and erosions within the mucous membranes. Patients with EBA have IgG autoantibodies directed against type VII collagen, a component of the anchoring fibrils of the basement membrane. It has been shown that autoantibodies bind to the NC-1 domain of type VII collagen within the lamina dura. Some patients have associated inflammatory diseases with Crohn’s disease, ulcerative colitis, thyroiditis, psoriasis, hepatitis C infection, diabetes mellitus and rheumatoid arthritis (occurring most frequently) [72,73]. Clinically, it can present with numerous phenotypes, though the most common are the mechanobullous and inflammatory variants. The first one (classic presentation) results in a lesion of the basement membrane with little inflammation; the second includes a significant infiltration of neutrophils [72,74]. Inflammatory variants of EBA resemble other autoimmune bullous dermatoses and include bullous pemphigoid-like (BP) EBA, mucosal pemphigoid-like (MMP) EBA, and linear IgA bullous dermatosis (LABD) EBA. BP-like EBA is characterized by extensive, tense blisters and erythematous blisters, usually involving the torso and extremities. The oral mucosa may be involved in approximately 50% of the cases, with erosions or intact oral vesicles. They can be hemorrhagic and can result in erosions, crusts and scars [75,76]. MMP-like EBA is characterized by predominant involvement of the mucosa, with lesions and scarring most often in the oral cavity; eye involvement leading to blindness has also been reported [53]. Diagnosis is established by clinical presentation and detection of autoantibodies bound to the basement membrane zone and against collagen VII in the serum [77,78]. A perilesional biopsy should be taken for direct immunofluorescence (DIF) microscopy, which would show linear deposits of IgG along the epidermal basement membrane zone. Less commonly, linear deposits of C3, IgA, or IgM may also be detected [79,80].

The treatment of EBA includes systemic corticosteroids such as colchicine, dap-sone, methotrexate, azathioprine, cyclosporine, mycophenolate mofetil, and cyclophosphamide. In addition, IVIG, rituximab, plasmapheresis and immunoadsorption and extracorporeal photochemotherapy may also be used. Early diagnosis and appropriate treatment are the basis for a good prognosis in EBA, which is a challenge because of its rarity and therefore the lack of randomized controlled trials [81].

### 5.5. Hyper-IgE Syndrome

Hyper-IgE syndrome (HIES), also called Job’s syndrome or Buckley’s syndrome, was first described as a rare primary immunodeficiency disease characterized by recurrent staphylococcal “cold” skin and pulmonary abscesses, eczematoid dermatitis, markedly elevated levels of serum IgE and eosinophilia, reduced neutrophil chemotaxis, and variably impaired T cell function. Since the initial descriptions of HIES, it has become abundantly evident that HIES is a multi-system disorder [80,82]. Individuals often present a common characteristic facial appearance with asymmetry, a fleshy nasal tip, deep-set eyes, and a prominent forehead. Typically, hyper IgE syndrome is characterized by the triad of eczema, recurrent skin and lung infections, and high serum levels of IgE [83,84]. In the oral cavity, HIES manifests with retained primary dentition and variations in oral mucosa and gingiva. Oral candidiasis (pseudomembranous, erythematous, median rhomboid glossitis, and angular cheilitis) is also common [85]. Intraoral lesions may be asymptomatic and require no intervention. These oral characteristics manifest earlier than the characteristic facial features, highlighting the potential role of oral phenotypes in early diagnosis [86]. Abnormalities in the oral cavity include the oral mucosa within gums, hard palate, the buccal mucosa, lips, and the dorsum of the tongue. Pathological lesions have been found in more than 75% of patients [87]. Most patients with HIES have palatal lesions with linear midline fibrosis, sometimes surrounded by clefts or grooves. The most significant for the tongue is a deep medial cleft in front of the circumvallate papillae, which may or may not have an overlying tissue flap. Even more common are lesions on the tongue consisting of grooves of varying depth that may be located or spread over the entire surface of the tongue. On the lips and cheeks, the lesions within the mucosa consist of superficial fissures and keratotic scales or plaques, sometimes resembling lichen planus. Oral lesions may be developmental, reactive changes resulting from chronic infections associated with the syndrome, or reflect the role of the HIES gene, STAT3, in epithelial development [87,88,89]. About 64% of patients with HIES show no exfoliation of the primary teeth, often preventing eruption of the permanent teeth [88,89,90,91]. This retention of the deciduous teeth may predispose to malocclusion. The abnormal persistence of Hertwig’s epithelial root sheath in primary teeth has been suggested as the cause of the lack of exfoliation, but the actual cause of the problem remains to be clarified [91]. Significant advances in the molecular understanding of both the clinical phenotype and the pathogenesis of the autosomal dominant form of HIES (AD-HIES) have been made recently. Although mutations in the cytokinesis 8 (DOCK8) gene dedicator are responsible for the autosomal recessive form of HIES, the more common and better-characterized form of HIES is caused by STAT3 mutations [89]. In the diagnosis of HIES, the NIH scoring system, based on the presence and severity of 21 clinical and laboratory results, was specifically developed [84].

At HIESS, therapy focuses on reducing bacterial and fungal infections with antimicrobial agents. However, gene therapies and adaptive cell therapies appear promising. For example, in the case of DOCK8 deficiency, hematopoietic stem cell transplantation (HSCT) may yield good results. This disease is very rare and its prognosis is not very good. It can cause high mortality and high susceptibility to the development of infections in patients. The major difficulty in this disease remains the lack of specific treatment [90].

A summary of the main features of oral manifestation in selected diseases is shown in Table 2.

## 6. Discussion

The oral mucosa is the first line of defense of the human immune system. Typically, pathogenic viruses or bacteria attack the surfaces of the oral mucosa, fighting the local immune system and healthy microflora. Immunoglobulin A (IgA) is the main immune factor in saliva and regulates the homeostasis of the oral microflora [91]. Several studies have described the ability to secrete IgA to protect the body against viral infections with pathogens such as rotaviruses or influenza viruses [92]. Secretory IgA works to protect the mucosal epithelium. Nicolas Millet et al. discovered that mucosal IgA can prevent commensal dysbiosis of *Candida albicans* in the oral cavity. In the study, they found that IgA is the dominant antibody isotype in the mucosal immune system. Immune exclusion is the primary mechanism by which secretory IgA (sIgA) blocks microorganisms from attaching to mucosal epithelial cells, thereby preventing colonization, damage, and subsequent invasion. Oral fungal colonization upregulates adaptive host responses, including the upregulation of the immune network for IgA production. *C. albicans* colonization increased the total salivary and tissue IgA levels, thereby preventing adhesion and invasion of the fungus [93]. Chang and co-workers [94] in their study suggest that IgA in saliva may bind to highly pathogenic oral bacteria, control their pathogenicity and maintain normal metabolic activity and alveolar bone level.

In the course of many systemic diseases, the oral cavity is the place where pathological symptoms appear. Autoimmune diseases with oral manifestations are not common in the population studied. Since the initial manifestations of most of these diseases occur in the oral mucosa, an earlier diagnosis and proper therapeutic protocol will delay the dissemination of these lesions, greatly contributing to a better prognosis and quality of life of the patients [95]. Femiano et al. [96] reported five cases of patients with PV who were clinically diagnosed as possibly suffering from recurrent aphthous stomatitis. Patients rarely report the intraoral formation of vesicles or bullae, and these lesions can seldomly be identified by clinical examination because of their thin, friable roof. The significant involvement of the buccal, alveolar and palatal mucosae agrees with the reports of Scully et al. and Robinson et al. [97,98]. In autoimmune diseases, antibodies directed against a specific structure may be present, and their presence leads to abnormalities in the oral mucosa. Changes in the amount or function of antibodies also lead to pathological symptoms in the oral cavity, not only in autoimmune diseases. When the level of antibodies is reduced, pathological microorganisms in the oral cavity develop and local homeostasis is disturbed. In MMP, Carey and Setterfield [52] found that serum samples are valuable for identifying the isotype, binding pattern, and titer of circulating antibodies. IgG antibodies are detected in 50–80% of patients and/or IgA antibodies in 60%. The authors showed that DIF was positive in 134/143 (93.7%) samples. In people with gingival MMP, biopsy of reflected alveolar mucosa was positive in 100% (17/17).

Sultan and co-workers [99], in their study, compared patients with oral MMP and PV who were diagnosed and treated at the Division of Oral Medicine and Dentistry, Brigham and Women’s Hospital, Boston and the Department of Dermatology, Massachusetts General Hospital, Boston. All MMP and PV cases showed the presence of IgG in the subepithelial and intraepithelial locations, respectively, in the DIF studies. The presence of IgA in addition to IgG was detected in three cases (12%) of MMP and in one case (3%) of PV. IgG-positive was found in 90% of PV cases, and IgG/IgA-positive in one case of PV (3%). Arisawa et al. [95] analyzed the main clinical and histopathological features of autoimmune diseases with oral manifestations such as mucous membrane pemphigoid (MMP) and pemphigus vulgaris (PV). Clinical signs of MMPs included ulcerated lesions (100%) with positive Nikolsky sign (50%). Visible blistering in approximately 33% of patients (painful in some cases) occurred. The alveolar mucosa was the most the affected area, then lips and cheeks equally, and soft palate mucosa. Histological analysis revealed the presence of subepithelial blisters and chronic inflammatory infiltrate with eosinophils in all cases. Epithelial atrophy and exocytosis were there observed in four microscopic samples.

In the study of Palmer and co-authors [100], they observed developed urticarial lesions with bullae in examined patients. Drug-induced LAD, like idiopathic LAD, can have a variety of presentations, including urticated plaques, erythema multiforme-like lesions and bullous eruptions. Mucosal involvement is frequent in idiopathic LAD, but it has been suggested that it is rare in the drug-induced form of the disease. They conducted a literature review which, compared to the cases, revealed that 13 of 29 patients (45%) had mucosal involvement.

Buckley et al. [101] found that recurrent staphylococcal abscesses and chronic eczema have been associated with extremely high serum concentrations of IgE. They also proved that the serum levels of other immunoglobulins (IgG, IgA, IgM and IgD) and the IgG subclasses in these patients were normal. Non-immune symptoms of HIES were found by Grimbacher [102] to extend to connective tissue abnormalities such as osteoporosis, fractures after minor trauma, overstretched joints and retention of primary teeth.

In idiopathic LABD, LAD285, BP180 and BP230 were identified as major target antigens. The soluble 120 kDA/97 kDa ectodomain is believed to contain the primary target IgG and IgA antibodies in bullous pemphigoid, LABD and other immunobullous diseases [69,73]. In drug-induced LABD, IgA antibodies against LAD285 and BP180, as in idiopathic LABDs, were identified [100]. Some drugs can cause an autoimmune response by cross-reacting with target epitopes and by exposing previously sequestered antigens to the immune system [103].

The mechanism of the immune deficiency of HIES remains poorly understood. There have been multiple reports with small numbers of patients and conflicting data as to what immune defect is present [104]. In laboratory tests, nearly all patients have significant increases in serum IgE (2000–100,000 IU/mL); IgE level begins to rise after birth and decline or even normalize in adulthood. Thus, a normal IgE level <2000 does not rule out the diagnosis. Eosinophilia occurs in more than 90% of patients (usually ≥700 mL^−1^ cells). Neither IgE level nor eosinophilia correlates with disease activity. Most HIES patients show a lower vaccine response due to abnormal B cell maturation, as exemplified by a reduced CD27+ switched memory B cell count. With the exception of a reduced central memory T cell count, T cell numbers and functions are usually normal [80].

Autoimmune diseases with symptoms in the oral cavity are not common. The initial symptoms within oral mucosa may lead to a proper diagnosis and treatment protocol and will delay the spread of these changes, contributing significantly to better prognosis and patient’s quality of life.

The main limitation of research on oral immunoglobulin disorders is the clinical oral examination (COE). The disorders we have described have symptoms occurring in the oral cavity, head and neck, making COE the main strategy used to detect them. A proper COE involves a thorough examination of the head and neck, evaluation of the oral mucosa by visual inspection under incandescent or halogen lighting, and palpation [105]. However, some conditions give a similar clinical picture by which the diagnosis can be erroneously made without additional, more detailed examinations. The small number of occurrences of immunoglobulin disorders with oral manifestations may indicate not only the lack of prevalence of these diseases in the population, but also the lack of diagnostics performed for them.

## 7. Conclusions

The low level of immunoglobulin in the oral cavity disrupts local homeostasis and enables the multiplication of pathological microorganisms. Patients with decreased immune function are predisposed to some systemic and oral manifestations, including inflammation of the periodontium and surrounding tissues. Immunoglobulin disorders with oral manifestations are not common, but knowledge about differentiation or diagnosis are essential for proper treatment. Early detection of autoimmune diseases, sometimes showing a severe evolution (e.g., PV), allows control of their dissemination to the skin and other body organs.

## Figures and Tables

**Figure 1 jcm-11-04873-f001:**
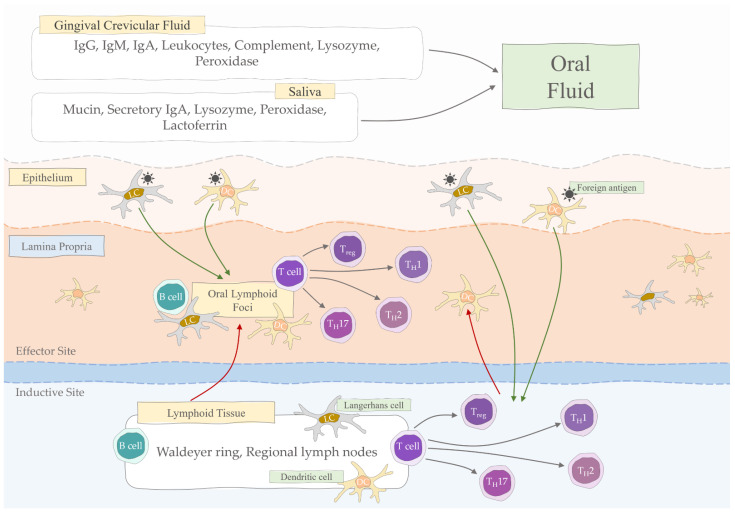
Oral mucosal immunity (Treg, regulatory T cells; TH, helper T cells; DC, dendritic cell; LC, Langerhans cell; IgA, immunoglobulin A; IgG, immunoglobulin G; IgM, immunoglobulin M).

**Table 1 jcm-11-04873-t001:** Characteristics of selected diseases manifested in the oral cavity.

Disease	Clinical Manifestations	Diagnostic Tests	Treatment
mucous membrane pemphigoid (MMP)	desquamative gingivitis, erythematous plaques and erosions covered by pseudomembranes located on the gingiva, palate, lips, tongue and cheek mucosa	clinical, histological and immunopathological findings, direct immunofluorescence (DIF) for immunoglobulin G, A and M (IgG, IgA, IgM) or complement component 3 (C3)	systemic corticosteroids (prednisone), immunosuppressive drugs (cyclophosphamide, azathioprine)
Pemphigus vulgaris (PV)	painful mucosal ulceration	immunoblotting with immunoprecipitation	systemic corticosteroids
Linear IgA bullous dermatosis (LABD)	blisters and erosions of oral mucosa, desquamative gingivitis	biopsies for histopathological examination, indirect immunofluorescence (IIF)	dapsone, an immunomodulatory sulphone
Epidermolysis Bullosa Aquisita (EBA)	vesicle and bullae formation on the skin and erosions within the mucous membranes	detection of autoantibodies bound to the basement membrane zone and against collagen VII in the serum, biopsy (direct immunofluorescence)	non-specific immunosuppression
Hyper-IgE syndrome (HIES)	skin and pulmonary abscesses, eczematoid dermatitis, markedly elevated levels of serum IgE and eosinophilia, reduced neutrophil chemotaxis	the National Institutes of Health (NIH) scoring system	antibiotic prophylaxis with trimethoprim-sulfamethoxazole

IgG: immunoglobulin G; IgA: immunoglobulin A; IgM: immunoglobulin M; C3: complement component 3; NIH: The National Institutes of Health.

**Table 2 jcm-11-04873-t002:** Summary of key features of oral manifestations in selected diseases.

Disease	Type	Immune Factors	Risk Factors	Symptoms
MMP	Autoimmune dermatoses disease	Linear deposition of IgG, IgA, C3, rarely IgG Target antigens in the basement membrane region of the epithelium: BP230, BP180, laminin 5, laminin 6, type VII collagen and β4	Age > 50 years, female sex	Subepidermal bullous lesions
PV	Autoimmune disease	Autoantibodies react to Dsg 1 and 3	-	Ulceration of the oral and esophageal mucosa, followed by skin involvement
LABD	Subepidermal vesiculobullous disease	Deposition of IgA rather than IgG in the basement membrane	Genetic factors, medication (vancomycin) and antibiotic use (cephalosporins, penicillins)	Oral lesions clinically identical to MMPs
EBA	Autoimmune blistering disease	IgG autoantibodies directed against type VII collagen	-	Blisters and blisters on the skin and erosions on the mucous membranes
HIES	Multisystem disorder	-	STAT3 mutations	Facial asymmetry, fleshy nose tip, deep-set eyes, prominent forehead, triad of eczema, recurrent skin and lung infections, high serum IgE levels, preserved primary dentition, oral mucosal and gingival lesions, oral candidiasis

IgG: immunoglobulin G; IgA: immunoglobulin A; IgE: immunoglobulin E; C3: complement component 3; MMP: mucous membrane pemphigoid; PV: pemphigus vulgaris; LABD: linear IgA bullous dermatosis; EBA: Epidermolysis Bullosa Aquisita; HIES: Hyper-IgE syndrome; STAT3: Signal transducer and activator of transcription 3.

## Data Availability

Not applicable.

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
