# Peer review of "Immunoglobulin Disorders and the Oral Cavity: A Narrative Review"

_jcm, 2022, doi:10.3390/jcm11164873_

Round 1

Reviewer 1 Report

The topic of the manuscript is the literature review on the potential relationships between immunoglobulin disorders and their oral manifestations.

The title and the abstract are informative. However, the title seems to be too general. The main text discusses the relevant references, including the recent ones from 2020 to 2022. The Conclusions seem to be “take-home” messages.

Moreover, the selected Special Issue “Clinical Updates in Endodontics” does not seem at all suitable for this manuscript.

Some following points must be clarified/corrected for the further processing of this manuscript.

Merits-related comments:

1.       The title should be modified and supplemented by the phrase “narrative review”.

2.       The Introduction section requires adding proper references (for example for GCF: e. g. 10.3390/molecules26051208, 10.3390/ijms22115463). Please also supplement the other missing references. It is suggested that the Introduction should mention possible autoimmune diseases predisposing to periodontitis or oral lesions (e. g. 10.3390/diagnostics9030077, 10.3390/ijerph182111521, 10.3390/jcm10091957, 10.3390/ijerph16050771, 10.3390/medicina58050621).

3.       In the introductory sections on salivary immunoglobulins and oral immunity, it is useful to consider the latest research on salivary immunoglobulin A in potentially autoimmune inflammatory bowel diseases (other than diagnostic use, interesting changes due to the biologic treatment of these patients): 10.3390/life11090943, 10.3390/life11121409. Among the factors mentioned affecting oral immunity (in lines 144-145), one of the most important, namely the level of oral hygiene, has been overlooked (proposed references: 10.3390/nu11122898).

4.       Although the manuscript does not constitute a systematic review, it would be good to add a new paragraph on the search and selection of the references included in the review, using part of the PRISMA guidelines (databases, keywords, timeframe etc.).

5.       Moreover, the main text should be enriched with clinical images of oral manifestations or summary tables, which would make it easier for readers to perceive and understand the review.

6.       Also, the potential limitations of the study could be explained in the new paragraph at the end of the Discussion.

Author Response

Dear Reviewer,

thank you for reviewing our manuscript titled: “Immunoglobulin disorders and the oral cavity “ for Journal of Clinical Medicine. We appreciate the time and effort that you have dedicated to providing your valuable feedback on our manuscript.  We have been able to incorporate changes to reflect most of the suggestions provided by you. We have highlighted the changes within the manuscript.

Here is a point-by-point response to your comments and concerns.

Merits-related comments:

The title should be modified and supplemented by the phrase “narrative review”.

RE: Thank you for this opinion. The title has been corrected as suggested by the Reviewer.

The Introduction section requires adding proper references (for example for GCF: e. g. 10.3390/molecules26051208, 10.3390/ijms22115463). Please also supplement the other missing references. It is suggested that the Introduction should mention possible autoimmune diseases predisposing to periodontitis or oral lesions (e. g. 10.3390/diagnostics9030077, 10.3390/ijerph182111521, 10.3390/jcm10091957, 10.3390/ijerph16050771, 10.3390/medicina58050621).

RE: Thank you for this opinion. We supplemented the manuscript with the missing references and added information on possible autoimmune diseases predisposing to periodontitis or oral lesions.

In the introductory sections on salivary immunoglobulins and oral immunity, it is useful to consider the latest research on salivary immunoglobulin A in potentially autoimmune inflammatory bowel diseases (other than diagnostic use, interesting changes due to the biologic treatment of these patients): 10.3390/life11090943, 10.3390/life11121409. Among the factors mentioned affecting oral immunity (in lines 144-145), one of the most important, namely the level of oral hygiene, has been overlooked (proposed references: 10.3390/nu11122898).

RE: Thank you for highlighting these important aspects. The information has been supplemented as suggested.

Although the manuscript does not constitute a systematic review, it would be good to add a new paragraph on the search and selection of the references included in the review, using part of the PRISMA guidelines (databases, keywords, timeframe etc.).

RE: Thank you for your remark, however, we believe that describing the search and selection of bibliography in a new paragraph is pointless. This is not a systematic review, which is characterized by the search for references through the use of specific keywords. In our publication, we used numerous formulations to find information of interest sometimes ignoring the time frame, as some of the studies were not repeated.

Moreover, the main text should be enriched with clinical images of oral manifestations or summary tables, which would make it easier for readers to perceive and understand the review.

RE: The text has been supplemented by a table summarizing aspects of the diseases we selected.

Also, the potential limitations of the study could be explained in the new paragraph at the end of the Discussion.

RE: We have added a short paragraph on the limitations that diagnosis of oral immunoglobulin disorders has.

Again, thank you for investing the time and effort to read and review our paper. We hope is is improved in the present form.

Best regards,

Paulina Niedźwiedzka-Rystwej

Reviewer 2 Report

I thought this was a comprehensive review of several disorders.  It is dense reading but that is likely related to the complexity of the discussed diseases.  

The content was interesting but out of my range of expertise so it is hard for me to comment on its accuracy.

While well written it was hard to get through and I wonder if there are some ways the authors could make the material easier for those less familiar with these disorders.

For example,  I think a list of the multiple abbreviations used would be helpful.  A summary table of the diseases would be helpful with pathophysiology, clinical manifestations, diagnostic tests, prognosis and treatment

I also think the authors should include brief descriptions  of  prognosis and treatment, eg. self-limiting, treat with steroids, immunosuppressants.

If possible, the diagnostic approach should be emphasized more since this is helpful for clinicians. 

I like the diagram but the acronyms should be in the key.

Author Response

Dear Reviewer,

thank you for giving us the opportunity to revise our manuscript titled: “Immunoglobulin disorders and the oral cavity “ to Journal of Clinical Medicine. We appreciate the time and effort that you ahave dedicated to providing your valuable feedback on our manuscript. We have been able to incorporate changes to reflect most of the suggestions provided by you. We have highlighted the changes within the manuscript.

Here is a point-by-point response to your comments and concerns.

For example, I think a list of the multiple abbreviations used would be helpful.

RE: We have added a list of abbreviations used in the review at the end of the manuscript.

A summary table of the diseases would be helpful with pathophysiology, clinical manifestations, diagnostic tests, prognosis and treatment

RE: We have added a table in the chapter on diseases. We limited it to clinical symptoms, tests and treatment for the reason of better readability of the table. We recognized that pathophysiology is described in detail in the text and reducing it to a table may contribute to misunderstanding the entire context.

I also think the authors should include brief descriptions of prognosis and treatment, eg. self-limiting, treat with steroids, immunosuppressants.

RE: Brief descriptions of prognosis and treatment have been completed as suggested.

If possible, the diagnostic approach should be emphasized more since this is helpful for clinicians.

RE: Thank you for this opinion. We present the diagnostics targets in all immunoglobulin disorders described in this work, such as Pemphigus vulgaris (PV), Mucous membrane pemphigoid (MMP), Linear IgA bullous dermatosis (LABD), Epidermolysis Bullosa Aquisita (EBA ), Hyper-IgE Syn-drome (HIES)  (in subsections 5.1 – 5.5). The text related to diagnostic approach was marked in manuscript in green. We hope that the manuscript is clinician-friendly, because Authors tried to put a great emphasis on the description of diagnosis process and differentiation of the oral cavity disorders.

I like the diagram but the acronyms should be in the key.

RE: We have added the acronyms to the description of the diagram.

Again, we would like to thank you for your time, expertise, and effort in correcting our paper. We hope that now it fulfills the requirements to be published in Journal of Clinical Medicine.

Kind regards,

Paulina Niedźwiedzka-Rystwej

Round 2

Reviewer 1 Report

The Authors have referred to all the comments of the Reviewers and improved the very good manuscript considerably. I have no further comments.

Only in line 66, the error appears - the word “salivary” should be instead of “serum”.